# An Improved RandLa-Net Algorithm Incorporated with NDT for Automatic Classification and Extraction of Raw Point Cloud Data

**Zhongli Ma, Jiadi Li \*, Jiajia Liu \*, Yuehan Zeng, Yi Wan and Jinyu Zhang**

School of Automation, Chengdu University of Information Technology, Chengdu 610225, China
\* Correspondence: lijiadi96@163.com (J.L.); liujj@cuit.edu.cn (J.L.)

**Abstract:** A high-definition map of the autonomous driving system was built with the target points of interest, which were extracted from a large amount of unordered raw point cloud data obtained by Lidar. In order to better obtain the target points of interest, this paper proposes an improved RandLa-Net algorithm incorporated with NDT registration, which can be used to automatically classify and extract large-scale raw point clouds. First, based on the NDT registration algorithm, the frame-by-frame raw point cloud data were converted into a point cloud global map; then, the RandLa-Net network combined random sampling with a local feature sampler is used to classify discrete points in the point cloud map point by point. Finally, the corresponding point cloud data were extracted for the labels of interest through numpy indexing. Experiments on public datasets senmatic3D and senmatickitti show that the method has excellent accuracy and processing speed for the classification and extraction of large-scale point cloud data acquired by Lidar.

**Keywords:** NDT registration; map building; RandLa-Net; random sampling; semantic segmentation

## 1. Introduction

Lidar is an application system that integrates laser technology, global positioning system technology, and inertial measurement unit (IMU) technology, and it is also an important sensor for environmental perception in autonomous driving systems. The proper processing of the collected raw point cloud data can provide important data for a high-definition map, and furthermore, as a scarce resource and just-needed product in the field of autonomous driving, high-definition road maps also play a central role in its entire field, which helps cars perceive complex road information (such as slope, curvature, heading, etc.) in advance, and to make the right decisions combine intelligent path planning [1].

Using the point cloud data obtained by the vehicle laser to build high-definition road maps mainly includes three steps: data acquisition, point cloud data processing and drawing. Among them, point cloud data processing is the key step to ensure the accuracy and quality of the point cloud. The traditional method firstly maps the road surface according to discriminative point cloud mapping features, and then manually labeled as point clouds for classification. Some scene elements and blind areas that cannot be recognized and scanned by point cloud classification algorithm still need to be accounted for by additional manual mining. In short, traditional methods have shortcomings, such as needing a long production cycle due to a large amount of control and measurement tasks, and thus they cannot meet the needs of making high-definition maps very well. Traditional methods also have difficulty in identifying path elements when dealing with some road sections [2].

An important purpose of raw point cloud data classification and extraction research is to quickly and accurately distinguish points in roads, cars, people, traffic signs and classes of interest; however, most of the defects of traditional methods come from manual point

cloud classification. In response to this problem, many scholars have begun to pay attention to the research on the automatic classification and extraction of point clouds, which can greatly improve the timeliness of processing point cloud data when making high-definition maps. Deep learning has shown excellent performance in object classification, extraction and recognition in computer vision, but it cannot directly deal with such discrete and irregular point cloud data and it can only process the point cloud data frame by frame. Based on the above description, there will be some problems as follows:

(1) The quantity of point cloud data is very large. Current point cloud semantic segmentation algorithms take a long time to train on large-scale data sets.

(2) In order to quickly obtain complete road sections, trees and other components to make high-definition maps, most point cloud semantic segmentation algorithms segment data frame by frame to reduce the amount of computation. Therefore, it is necessary to construct the original data into a complete point cloud map, and then perform semantic segmentation. In this way, the complete information of the road section can be obtained.

(3) The point cloud semantic segmentation is designed to assign labels to each point and classify points, though it is impossible to obtain point cloud sets from unclassified point data. In other words, the region of interest cannot be directly extracted, it is necessary to make labels for the point clouds, and then extract the corresponding point cloud data in turn according to the corresponding index.

To solve the above problems, the aim of this paper was to find an automatic classification and extraction method for large-scale point cloud data, and our method can achieve the fast and efficient classification and extraction of a large number of original point cloud data for the construction of high-definition maps of autonomous driving systems. Furthermore, our method can also provide accurate data. The contributions of our method are as follows:

(1) Based on analyzing and comparing the existing registration and semantic segmentation methods, we chose to integrate NDT registration into the RandLa semantic segmentation algorithm to process original point cloud data.

(2) We tested the two algorithms on the public datasets KITTI [3], Semantic3D [4], and SemanticKITTI [5], respectively, and we chose to fuse the two algorithms according to the experimental results.

(3) We give the description of the basic process of the improved RandLa-Net (network structure based on random sampling and local feature aggregation) algorithm incorporated with NDT, and use the improved method to perform many experiments on public datasets. The experimental results show that the data processed by our method can be directly used for the construction of a high-definition map.

## 2. Related Work

### 2.1. Methods for Point Cloud Data Registration and Semantic Segmentation

The traditional algorithms of point cloud registration are roughly divided into two categories: coarse registration and fine registration [6]. Coarse registration refers to the registration by calculating an approximate rotation and translation matrix between two point clouds. This method is generally used when the relative positional relationship between two point clouds is unknown. The fine registration refers to making the rotation and translation matrix more accurate by calculation when the rotation and translation matrix is known. Most of the point cloud information collected by vehicle radar is coarse registration, which mainly includes:

(1) A registration method based on local feature description. The point feature histograms (PFH) methods proposed by Rusu et al. [7] use point feature histograms to characterize the local geometry of 3D points for registration;

(2) Method based on probability distribution. The normal distributions transform (NDT) algorithm proposed by Biber [8] et al. is a rough registration method that uses range scanning first, that is, the normal distribution transformation is performed after point

cloud matching. In recent years, point cloud registration methods based on deep learning have also been widely proposed and applied. Aoki [9] et al. used PointNet to map the found feature points to a high-dimensional space, and then regarded the vector formed by each feature point as an image in the high-dimensional space, and finally used the traditional image registration algorithm (Lucas-Kanada, LK) [10] for point cloud registration. Wang [11] et al. extracted the features of the point cloud to be registered; they used the improved transformer network to merge the information between the point clouds, calculated the soft matching between the point clouds, and then used the differentiable singular value decomposition module to extract the rigid body changes for point clouds. Here, cloud registration [12] was combined with keypoint detection to solve the non-convexity and local registration problems of registration.

One type of method is to improve the traditional point cloud semantic segmentation algorithms, such as random sample consensus (RANSAC), density-based spatial clustering of applications with noise (DBSCAN), and region growth algorithm (region growing) [13–15]. These algorithms have high requirements for the quality of point cloud data and have low accuracy and slow speed when processing large-scale point cloud data. The other type of method is based on deep learning network, which mainly includes:

(1) Projection-based network [16]: due to the inhomogeneity of point cloud data, it is impossible to directly use the convolutional neural network on point cloud. To make use of two-dimensional convolutional neural network, the projection-based network chooses to project a three-dimensional point cloud onto a two-dimensional image and then input it into the network [17], but the projection process may lead to the loss of geometric information, and the method lacks the ability of non-local geometric features [18];

(2) Voxelization-based network [19]: for the disorder and irregularity of the point cloud, the disordered point cloud is voxelized into an ordered voxel block, and then a three-dimensional convolutional neural network is used to process the ordered voxel block. The main limitation of the method is that the computational cost is too high, especially when dealing with large-scale point clouds. It cannot meet practical applications [20], and the volume setting of the voxel block will affect the final segmentation effect. Furthermore, due to the sparsity of the point cloud, there will be empty voxel blocks generated, wasting the computational cost [21];

(3) Network based on the neural architecture: Hu Q [22] et al. designed an efficient neural architecture network structure based on random sampling and local feature aggregation (RandLa-Net). It can directly process large-scale point cloud data without any preprocessing.

### 2.2. Data Format Requirements in Unmanned High-Definition Map Construction

In order to make the high-definition electric urban map, a designed unmanned vehicle (shown in Figure 1) equipped with two 32-line lidars was used to collect the point cloud data of some road sections, and the sampling frequency of this lidar is 10 Hz. In order to restore the real road, we need to extract the feature information of the road from the large amount of point cloud data collected by this lidar, such as roads, vehicles, trees, buildings and pedestrians.

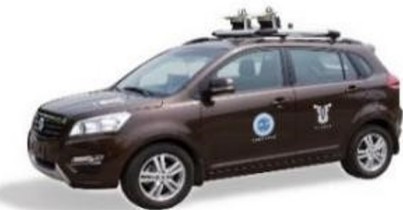

**Figure 1.** Unmanned vehicle with two 32-line lidars.

The construction of high-definition maps requires the classification of road point features by class and the three-dimensional coordinates of each point. The coordinates are listed in $(x, y, z)$ format, and the coordinate system is based on the starting position of the driverless car as the origin.

Because there are a lot of complex targets in the point clouds and the number of data collected is large, not all the collected road section information have a unified initial position, so the registration method based on the deep learning registration method is difficult and time-consuming to train. At the same time, although the RandLa-Net algorithm can effectively segment the point cloud map, the format of the single frame data output is '.label', which cannot be directly used to build the high-definition electric urban map.

Based on the above analysis, we integrated the traditional mature NDT registration algorithm into the RandLa-Net algorithm, so that the results of the semantic segmentation are more suitable for the construction of high-definition maps.

## 3. Creation of Global Map of Point Cloud Based on NDT

### 3.1. Registration Algorithm Based on NDT

NDT is normal distribution transformation. After gridding the reference point, the normal distribution transformation is performed one by one to complete the modeling of all reconstructed points. The specific operations are as follows.

First, the point cloud space is divided into cells with the same size according to certain rules.

Then, the following actions are performed on each cell:

Step1. Collect all points contained in this box: $X_i = 1 \ldots n$;

Step2. Calculate the average:

$$q = \frac{1}{n} \sum_i X_i \tag{1}$$

Step3. Calculate the covariance matrix:

$$\Sigma = \frac{1}{n} \sum_i (X_i - q)(X_i - q)^t \tag{2}$$

Step4. The probability of measuring a sample at point $x$ contained in this cell is now modeled by the normal distribution N $(q, \Sigma)$:

$$P(x) \sim exp\left(-\frac{(x-q)^t \Sigma^{-1}(x-q)}{2}\right) \tag{3}$$

Figure 2 shows the effect after meshing the 3D point cloud, the original frame and the visualization after NDT. The visualization is created by evaluating the probability density of each point, with bright areas indicating high probability density. Then, the normal distribution transformation is used for the registration between the two frames of point clouds, and the spatial mapping $T$ between the radar coordinate systems of the two frames of point clouds is given by:

$$T : \begin{matrix} x' \\ y' \end{matrix} = \begin{pmatrix} cos(\varphi) & -sin(\varphi) \\ sin(\varphi) & cos(\varphi) \end{pmatrix} \begin{pmatrix} x \\ y \end{pmatrix} + \begin{pmatrix} t_x \\ t_y \end{pmatrix} \tag{4}$$

where $(t_x, t_y)$ refers to the original position of the reference frame, $(x', y')$ is the position of the frame to be registered, $T$ describes the translation and rotation relationship between the two, $\varphi$ is the rotation angle, and $x$ and $y$ are the translation distances.

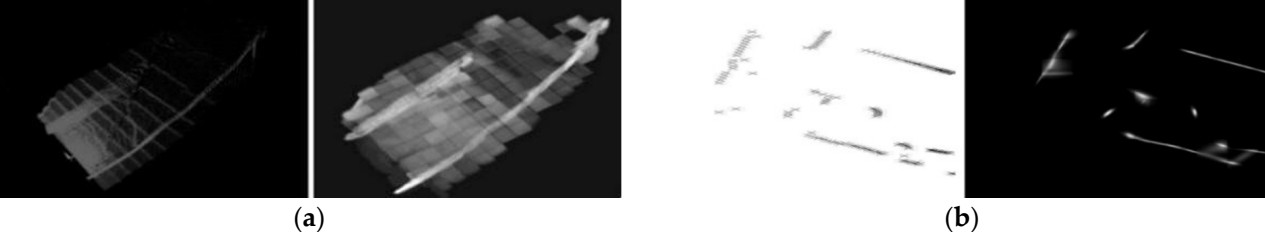

(**a**)                                                                                          (**b**)

**Figure 2.** The original data and the resulting probability density: (**a**) mesh the point cloud; and (**b**) the original scan and the resulting probability density.

The specific operations of registration are as follows:

Step1. Build the NDT based on the first frame scan;

Step2. Estimate initialization parameters;

Step3. For each sample to be registered, map the reconstructed point into the coordinate system of the reference frame according to the parameters;

Step4. Determine the corresponding normal distribution for each mapped point;

Step5. Determine the score of the parameter by evaluating the distribution of each mapped point and sum up the results;

Step6. Return to step 3 until the convergence criteria are met and the registration is completed.

The score is calculated as follows:

$$score(p) = \Sigma_i exp\left(\frac{-(x' - q_i)^t \Sigma_i^{-1}(x_i' - q_i)}{2}\right) \tag{5}$$

where $p$ is a vector of parameters to be estimated, $x_i$ is the point in the second frame of point cloud data, $x_i'$ is the point $x_i$ mapped to the coordinate system of the first frame point cloud data according to the parameter $p$, that is, $x_i' = T(x_i, p)$. $\Sigma_i$ and $q_i$ are the covariance matrix of the point cloud data in the first frame and the mean value of the normal distribution corresponding to the point $x_i'$. A mapping according to $p$ can be considered optimal if the sum of the normal distributions of all points $x_i'$ evaluated using the parameters $\Sigma_i$ and $q_i$ are at maximum, that is, the sum of the scores of $p$ is optimal.

The overall process of point cloud data registration based on NDT is shown in Figure 3.

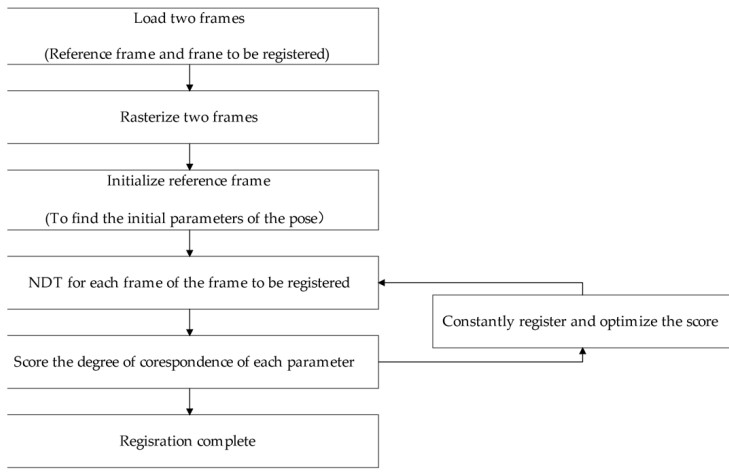

**Figure 3.** The flow of the NDT registration algorithm.

### 3.2. Point Cloud Map Creation Based on NDT

The point cloud map is created to avoid frame-by-frame processing for subsequent point cloud classification, and to improve the efficiency of the automatic classification of point cloud data. The main task of point cloud mapping is to use the collected point cloud data frame by frame to build a complete point cloud map. The algorithm flow is shown in Figure 4.

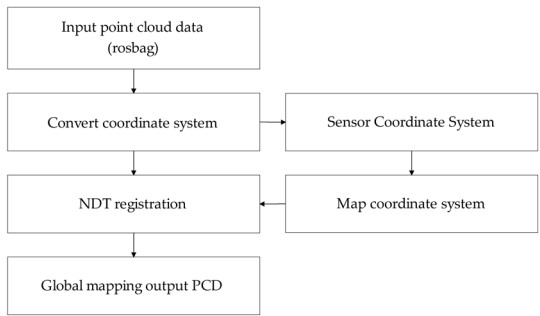

**Figure 4.** The process of point cloud mapping.

In Figure 4, the coordinate system transformation adopts the TF coordinate system transformation. If the coordinates of a point in the radar coordinate system are $P_L$ ($X_L$, $Y_L$), then the coordinates of a certain point in the map after conversion are $P_M$ ($x_m$, $y_m$), and the map coordinate system is a fixed coordinate system. The coordinate system is the same as the world coordinate system, then:

$$R \times P_L + t = P_M \tag{6}$$

where $R$ is the transformation matrix, so that the poses of the two coordinate systems are consistent.

$$R = \begin{bmatrix} cos(\theta) & -sin(\theta) \\ sin(\theta) & cos(\theta) \end{bmatrix} \tag{7}$$

In Formula (6), $t$ is the position where the origin of the sensor coordinate system is mapped to the map coordinate system. In general, the initial position of the sensor is the origin of the map, and $t = (x_o, y_o)$ can be set directly.

In Formula (7), $\theta$ is the heading angle during the driving process. According to the right-hand rule, the counterclockwise rotation around the z axis of the map coordinate system is positive, then $\theta$ can be directly brought into the transformation matrix, and the distance from the sensor coordinates to the map coordinates is transformed to:

$$P_M = \begin{pmatrix} cos(\theta) & -sin(\theta) \\ sin(\theta) & cos(\theta) \end{pmatrix} \times P_L + \begin{pmatrix} x_o \\ y_o \end{pmatrix} \tag{8}$$

The relationship between the sensor coordinate system and the map coordinate system is shown in Figure 5.

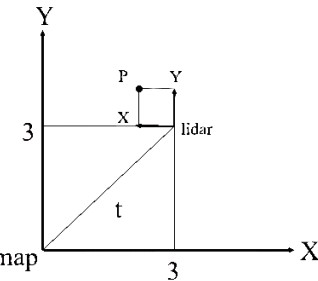

**Figure 5.** The relationship between the sensor coordinate and the map coordinate system.

### 3.3. Test of NDT Global Mapping

3.3.1. Dataset Selection

The KITTI dataset consists of point cloud data collected by 64-line 3D Lidar combined with two gray-scale cameras, two color cameras and four optical lenses, and the sampling frequency of Lidar is 10 Hz. The entire dataset consists of 389 pairs of stereo images and optical flow images, and more than 200 k 3D annotated objects. The KITTI dataset is often used in 3D object detection and point cloud segmentation, the part samples of dataset are shown in Figure 6.

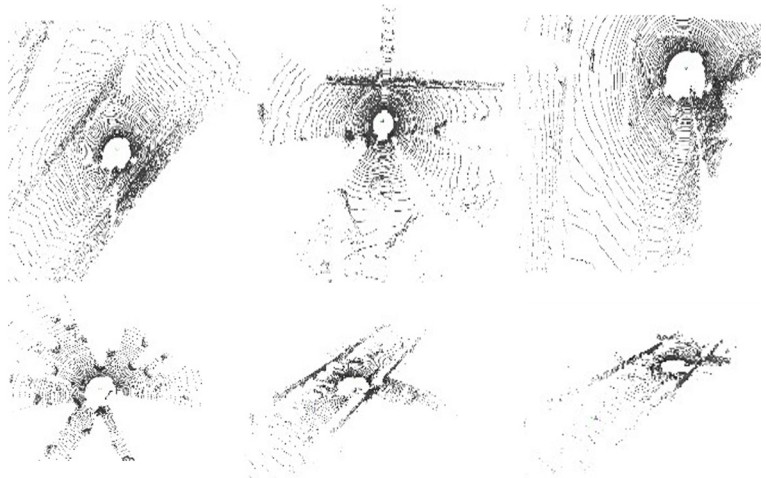

**Figure 6.** Partial image examples in the dataset.

3.3.2. Test Results

The CPU model of the computer is i7-10700k, the GPU model is NVIDIA RTX3090, and the memory size is 30 GB; the operating system is Ubuntu18.04, and the operating platform is Pycharm and PCL 1.10.0. The pseudocode for NDT registration is shown in Figure 7.

---

**Algorithm 1** Register scan X to reference scan Y using NDT

---

**Input:** $X = x_i \in X$; $Y = y_i \in Y$
**Output:** score(p)
 1: **function** M(y) Funs(X, Y, P):
 2:     Initialisation
 3:     allocate cell structure
 4:     **for** all points Y **do**
 5:         $find the cell b_i \in \beta\ that\ contains\ Y$
 6:         $store\ y_i\ in\ b_i$
 7:         $y' = y'_1, \dots, y'_n$
 8:         $\sum = \frac{1}{n} \sum_i (X_i - q)(X_i - q)^t$
 9:     **end for**
10:
11:     **while** not converged **do**
12:         score(p) = 0
13:         **for** all points X **do**
14:             $score\,(P) = score + \sum_i \exp(\frac{-(x_i - q_i)^t \sum_i^{-1} (x_i - q_i)}{2})$
15:         **end for**
16:         update score(P)
17:     **end while**
18:     **return** score(P)

---

**Figure 7.** Pseudocode for NDT registration algorithm.

The point cloud global mapping is performed on the test sets 11–21 of KITTI and the results are shown in Figure 8, where we can see that the point cloud global mapping is very densely reconstructed by NDT algorithm, and we can clearly see the vehicles on the road and the trees and buildings on both sides, even in the unsegmented state. In addition, due to the NDT registration using a one-time initialization, the process of algorithm execution does not need to consume much computing power to calculate the nearest search matching point, and the registration only takes 0.18 s per meter distance.

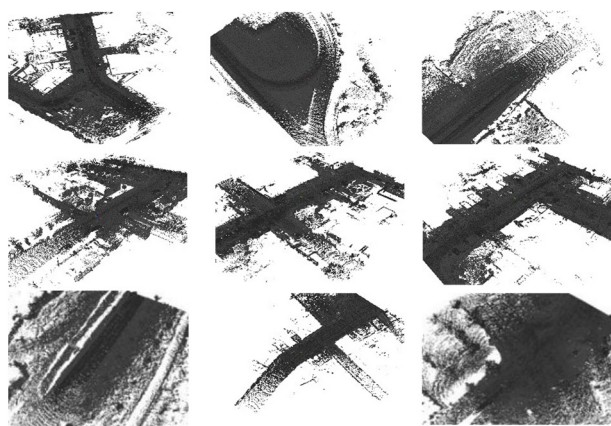

**Figure 8.** The effect of point cloud mapping.

## 4. Point Cloud Semantic Segmentation Based on RandLa-Net

*4.1. Point Cloud Semantic Segmentation Algorithm Based on RandLa-Net*

Point cloud semantic segmentation is to add semantic labels for each point, and to classify point clouds into different point subsets, and to make sure the same point cloud set has similar features, such as vehicles, roads, or pedestrians.

The semantic segmentation of a point cloud based on RandLa-Net: first, reduce the density and computational cost of the point cloud by random sampling, and then use the local feature aggregator to collect the features of the point cloud so as to avoid losing some important feature information of the point cloud due to random sampling. Finally, these features are aggregated and the point cloud is classified so that each point has corresponding label information. The specific segmentation process is shown in Figure 9.

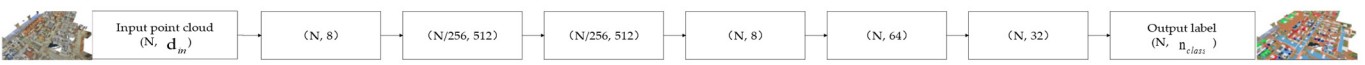

**Figure 9.** The process of semantic segmentation.

In Figure 9, (N, D) represent the point cloud number and feature dimension, respectively. FC represents the fully connected layer, LFA represents local feature aggregation, RS represents the random sampling, MLP represents shared multilayer perceptron and US represents upsampling.

### 4.1.1. Random Sampling

RS is to take n points from N points as samples, where each point has the same probability of being selected, and there is no special correlation between any two points. Its computational complexity is O (1). Compared with farthest point sampling and inverse density importance sampling, random sampling is the most computationally efficient, which only takes 0.004 s to process 106 points [22].

To evaluate the sampling efficiency of common types of samplings including farthest point sampling (FPS), inverse density importance sampling (IDIS), random sampling (RS), generator-based sampling (GS), continuous relaxation-based sampling (CRS) and policy gradient-based sampling (PGS), each of the above sampling methods is tested with point

cloud data with the numbers of 103, 104, 105 and 106 in turn. Point cloud data generally need to be downsampled five times, and each downsampling on a single GPU only retains 1/4 of the original points. The time and memory consumption of each sampling method are compared, and the results are shown in Figure 10 [23], the dashed lines represent the estimated value of the memory consumption.

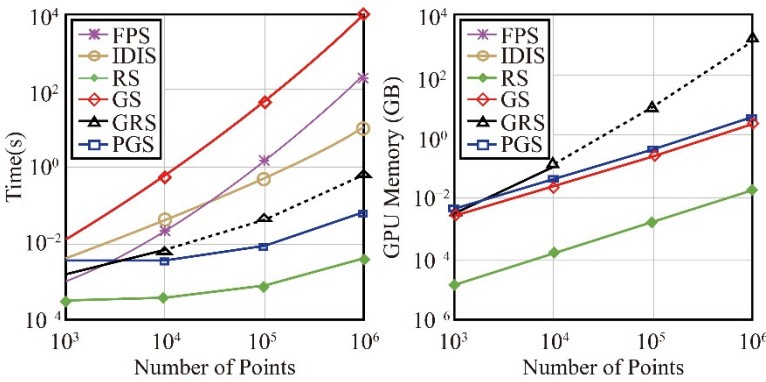

**Figure 10.** Efficiency comparison of different sampling methods.

4.1.2. Local Feature Aggregator

In order to retain the important feature information of the next point, the algorithm performs local feature aggregation after random sampling. This local feature aggregator is applied once at each point, which consists of three parts: (1) local spatial encoder (LocSE); (2) attention pooling layer; and (3) dilated residual block. The specific network structure is shown in Figure 11.

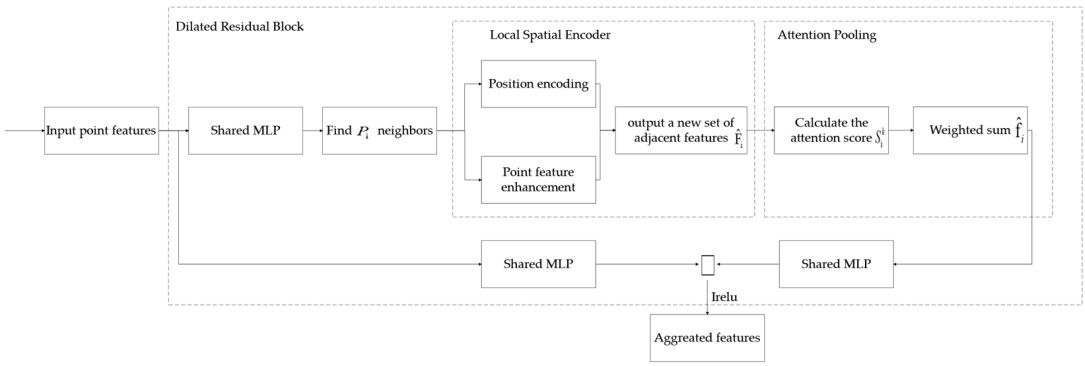

**Figure 11.** Local feature aggregator network structure.

(1)　Local Spatial Encoder

Given a point cloud P and the features of all points, this local space encoder stores the xyz coordinates of all adjacent points so that the features of the corresponding points also have corresponding coordinate positions. The local spatial encoder can also observe geometric patterns in blocks or regions, and the entire network can efficiently learn the complex local structure of point cloud data. Specific steps are as follows:

Step1. Find neighbors.

Step2. Position encoding of relative points. For each nearest point $\{p_i^1 \dots p_i^k \dots p_i^K\}$ of center point $p_i$, the corresponding positions are encoded as follows:

$$r_i^k = MLP\left( p_i \oplus p_i^k \oplus \left( p_i - p_i^k \right) \oplus \| p_i - p_i^k \| \right) \tag{9}$$

where $\oplus$ is the connection operation and $\| \cdot \|$ represents the calculation of the Euclidean distance between adjacent points and a given point $p$.

Step3. Point feature enhancement. For each adjacent point $p_i^k$, concatenate the encoded relative point position $r_i^k$ with its corresponding point feature $f_i^k$ to obtain an enhanced feature vector $\hat{f}_i^k$.

Step4. The output is a new set of adjacent features:

$$\hat{F}_i = \left\{ \hat{f}_i^1 \ldots \hat{f}_i^k \ldots \hat{f}_i^K \right\} \tag{10}$$

(2)　Attention Pooling Layer

This module is used to aggregate adjacent point features $\hat{F}_i$ and uses the attention mechanism to learn important local features spontaneously. The attention mechanism consists of the following parts:

Part1. Calculate the attention score. The formula is as follows:

$$s_i^k = g\left( \hat{f}_i^k, W \right) \tag{11}$$

where $g()$ represents a shared function to learn a unique attention score for each feature and $W$ is the learnable weight of the shared MLP.

Part2. The formula for the weighted summation is as follows:

$$\hat{f}_i = \sum_{K=1}^{K} \left( \hat{f}_i^k \cdot s_i^k \right) \tag{12}$$

(3)　Dilated Residual Block

Finally, in order to preserve an important feature information of all points before sampling as much as possible, this algorithm uses multiple local spatial encoders and expands the residual block formed in the stack of attention-eating layers and skip connections.

*4.2. Point Cloud Semantic Segmentation Test Based on RandLa-Net*

4.2.1. Dataset Selection

The experiment was conducted on the SemanticKITTI dataset. We selected the point cloud data from the first sequence to tenth sequence as the training set, and the point cloud data from the eleventh sequence to twenty-first sequence as the test set. Annotation categories are divided into: cars, bicycles, motorcycles, trucks, other vehicles, people, cyclists, motorcyclists, roads, parking lots, sidewalks, other surfaces, buildings, fences, vegetation, tree trunks, terrain, utility poles and traffic signs—19 categories in total, which are all important components in the autonomous driving traffic environment [5].

The Semantic3D dataset is a global point cloud image of different urban scenes obtained by static scanning with advanced equipment, and the dataset has more than 4 billion points in total. The annotation categories are artificial terrain; natural terrain; high vegetation; low vegetation; buildings; landscapes; objects; and cars—all of which are the main components of the urban environment [4].

4.2.2. Algorithm Testing

The CPU model of the test computer is i7-10700k, the GPU model is NVIDIA RTX3090 and the memory size is 30 GB; the operating system is Ubuntu18.04, the platform for deep learning uses the TensorFlow, the initial learning rate is set to 0.01 and it is reduced by 5% after each epoch. The number of closest points K is set to 16, the batch processed per iteration is set to 8 and a fixed number of points (approximately $10^5$) are sampled as input. The SemanticKITTI and Semantic3D datasets are used as the training and testing sets, respectively.

Table 1 shows the results of the training with different approaches on the semantickitti dataset. The average intersection ratio (mIoU) of all categories is used as a quasi-index, and the parameter column refers to the number of network parameters in the algorithm. It can

be seen that the mIoU of all categories obtained by RandLa-Net is significantly better than that of other algorithms, and RangeNet53++ has the best segmentation accuracy for small targets such as traffic signs and bicycles [24], which is because the network parameters of RangeNet53++ [24] are more than 40 times higher than those of RandLa-Net.

**Table 1.** Quantitative results of different approaches on semantickitti.

| Methods | PointNet | PointNet++ | SqueezeSeg | DarkNet21Seg | RangeNet53++ | RandLa-Net |
|---|---|---|---|---|---|---|
| mIoU(%) | 14.1 | 19.8 | 28.8 | 45.6 | 52.1 | 53.7 |
| parameter (M) | 3 | 6 | 1 | 25 | 50 | 1.24 |
| road | 61.6 | 71 | 85.3 | 91.4 | 91.8 | 91.7 |
| side-walk | 35.5 | 41.3 | 54.1 | 73 | 74.2 | 77.1 |
| parking | 15.6 | 18.3 | 26.9 | 56 | 63.9 | 41.2 |
| other-ground | 1.2 | 5.2 | 4.4 | 26.4 | 27.8 | 38.9 |
| building | 41.2 | 61.5 | 56.3 | 81.9 | 87.4 | 88.2 |
| car | 46.1 | 53.7 | 68.5 | 85.1 | 90.4 | 93.3 |
| truck | 0.1 | 0.9 | 3.3 | 18.1 | 24.7 | 40.1 |
| bicycle | 1.3 | 1.9 | 15 | 26.2 | 25.7 | 15.5 |
| motorcycle | 0.3 | 0.2 | 4.1 | 26.5 | 34.4 | 28.8 |
| other-vehicle | 0.7 | 0.2 | 3.5 | 15.6 | 22.9 | 38.5 |
| vegetation | 30 | 46.5 | 60 | 77.6 | 80.5 | 84.5 |
| trunk | 4.6 | 13.8 | 24.3 | 47.4 | 55.1 | 40.1 |
| terrain | 17.6 | 30 | 53.7 | 63.6 | 64.5 | 72.1 |
| person | 0.2 | 0.9 | 12.9 | 31.8 | 38.3 | 53.4 |
| bicyclist | 0.2 | 1 | 13.1 | 33.6 | 38.8 | 53.36 |
| motorcyclist | 0 | 0 | 0.9 | 4 | 4.8 | 7.2 |
| fence | 12.9 | 16.9 | 29.9 | 52.3 | 58.6 | 44.5 |
| pole | 2.4 | 6 | 17.8 | 36 | 47.9 | 51.3 |
| traffic sign | 3.7 | 8.9 | 24.5 | 50 | 55.9 | 38.6 |

Table 2 shows the train results of different approaches on Semantic3D, and the mean cross-over-union ratio (mIoU) and overall accuracy (OA) for all classes were used as standard metrics.

**Table 2.** Quantitative results of different approaches on Semantic3D.

| Methods | SnapNet | ShellNet | GACNet | KPConv | RandLa-Net |
|---|---|---|---|---|---|
| mIoU(%) | 59 | 69.1 | 70.7 | 74.6 | 77.4 |
| OA(%) | 88.6 | 91.8 | 94 | 92.9 | 94.8 |
| man-made terrain | 81 | 86.4 | 96.4 | 90.9 | 94.2 |
| natural terrain | 77.2 | 77.7 | 92.6 | 82.8 | 91.4 |
| high vegetation | 79.7 | 88.5 | 87.9 | 84.1 | 82.9 |
| low vegetation | 22.9 | 60.6 | 44 | 47.8 | 52 |
| buildings | 91.1 | 94.2 | 83.2 | 94.5 | 94.7 |
| hard scape | 18.4 | 37.3 | 31 | 40 | 54.9 |
| scanning artefacts | 37.3 | 43.5 | 63.5 | 77.3 | 70.9 |
| Cars | 64.4 | 77.8 | 76.2 | 79.7 | 76.9 |

RandLa-Net significantly outperforms other algorithms in mIoU and OA. The test accuracy of different methods for eight kinds of targets can be seen in Table 2, and the test accuracy of RandLa-Net algorithm is also better than most algorithms. The point cloud segmentation effect on different datasets is shown in Figure 12.

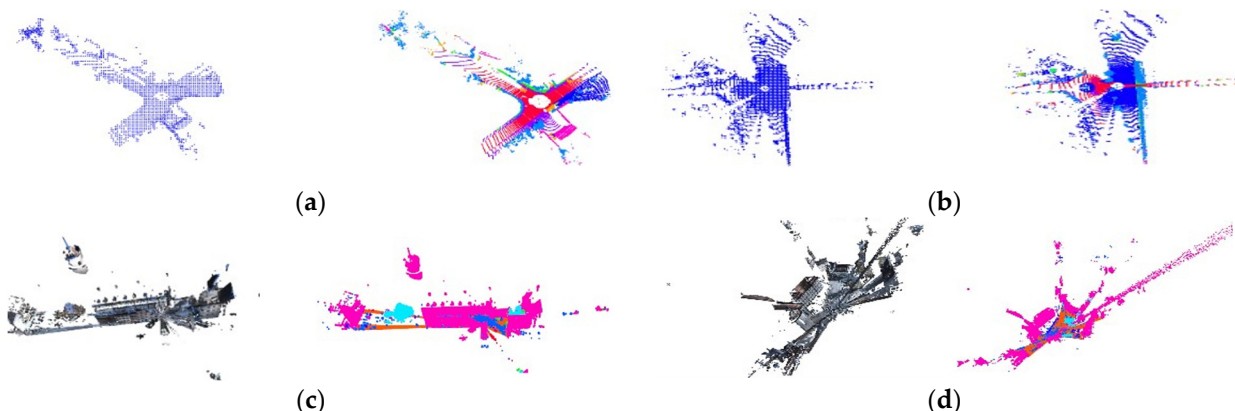

**Figure 12.** The segmentation effect on different datasets: (**a**) frame-by-frame segmentation effect of SemanticKITTI-11; (**b**) frame-by-frame segmentation effect of SemanticKITTI-12; (**c**) panoramic split effect of Semantic3D-1; and (**d**) panoramic split effect of Semantic3D-2.

Table 3 shows the total time and memory consumption of different methods. It can be seen in Table 3. RandLa-Net has the shortest processing time and the most maximum inference points. Although the number of network parameters of the SPG algorithm is the least, the processing time of point clouds is very long and the overall effect is not as good as RandLa-Net, due to the complex geometric division and hypergraph construction steps. Therefore, RandLa-Net is the most efficient network.

**Table 3.** Efficiency of the semantic segmentation of different methods on sequence 08.

|  | Total Time (s) | Parameters (Millions) | Maximum Inference Points (Millions) |
|---|---|---|---|
| PointNet | 192 | 0.8 | 0.49 |
| PointNet++ | 9831 | 0.97 | 0.98 |
| PointCNN | 8142 | 11 | 0.05 |
| SPG | 43584 | 0.25 | - |
| KPConv | 717 | 14.9 | 0.54 |
| RandLa-Net | 185 | 1.24 | 1.03 |

## 5. Comprehensive Test of Automatic Classification and Extraction of Raw Point Cloud Data

### 5.1. The Flow of the Algorithm

The improved RandLa-Net algorithm incorporated with NDT registration can directly process the complete point cloud map, and further obtain the data required by the high-definition map. The specific algorithm flow is shown in Figure 13.

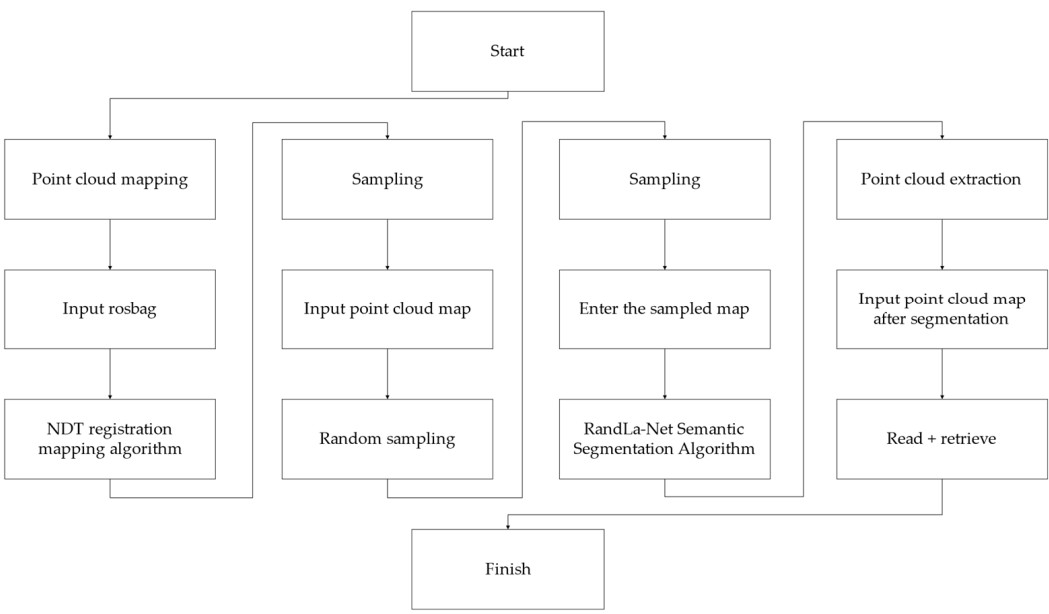

**Figure 13.** Process of the improved RandLa-Net algorithm.

*5.2. Test Data*

Take the SemanticKITTI Dataset 03 as an example for testing. Figure 14 shows the original frame-by-frame data of the 03 point cloud.

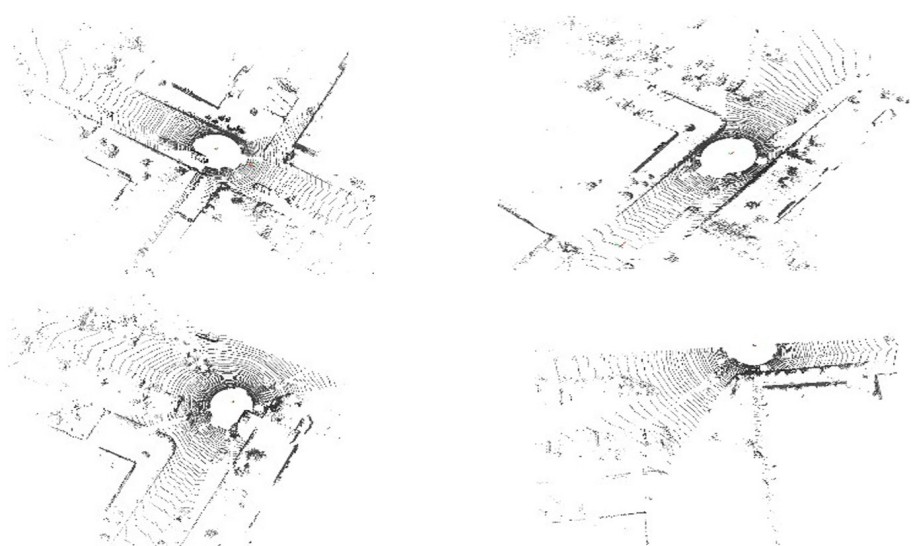

**Figure 14.** Original dataset.

*5.3. Point Cloud Mapping*

Firstly, load the rosbag of the original dataset, then use the NDT algorithm to perform the coordinate transformation and registration on the frame-by-frame point cloud data, and build a map globally. The output effect is shown in Figure 15.

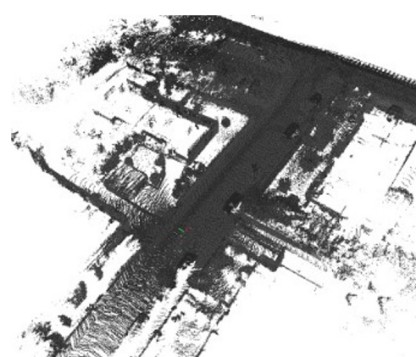

**Figure 15.** Point cloud mapping.

*5.4. Random Sampling*

In order to reduce the amount of computation, the random sampling of data is required before semantic segmentation, and the sampling effect is shown in Figure 16. We can see that the number of point clouds is significantly reduced and the features become blurred. However, the local feature aggregator of the subsequent RandLa-Net algorithm will solve this problem.

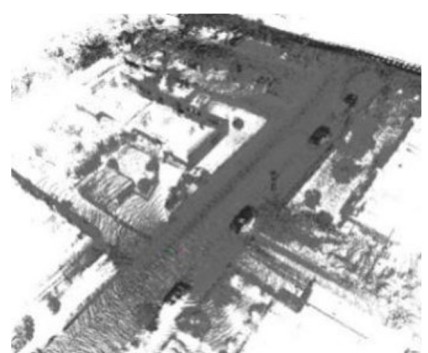

**Figure 16.** Random sampling.

*5.5. Point Cloud Semantic Segmentation*

Load the bin format file of the global point cloud image, and use the RandLa-Net algorithm to semantically segment the point cloud image. Figure 17a is the visualization of semantic segmentation on the dataset using our improved RandLa-Net algorithm, and Figure 17b is the visualization of semantic segmentation on the same dataset using the original RandLa-Net algorithm. Table 4 shows the specific effect of the test, which is reflected from the indicator mIoU. We can see that the segmentation effect does not decrease the accuracy due to the change from frame-by-frame segmentation to global segmentation.

Figure 17a is the visualization effect of the global point cloud map classification using RandLa-Net fused with NDT. Figure 17b is the visualization effect of the frame-by-frame point cloud classification using the original RandLa-Net network. Figure 17c is the label information corresponding to each color in the visualization. It can be seen that the algorithm in this paper can be used to classify the point cloud map at one time, and then the required area can be directly extracted. However, since the point cloud data processed by the original RandLa-Net algorithm was not registered and then there are a large number of redundancy points so that the point cloud of each road element cannot be directly extracted for making high-definition maps.

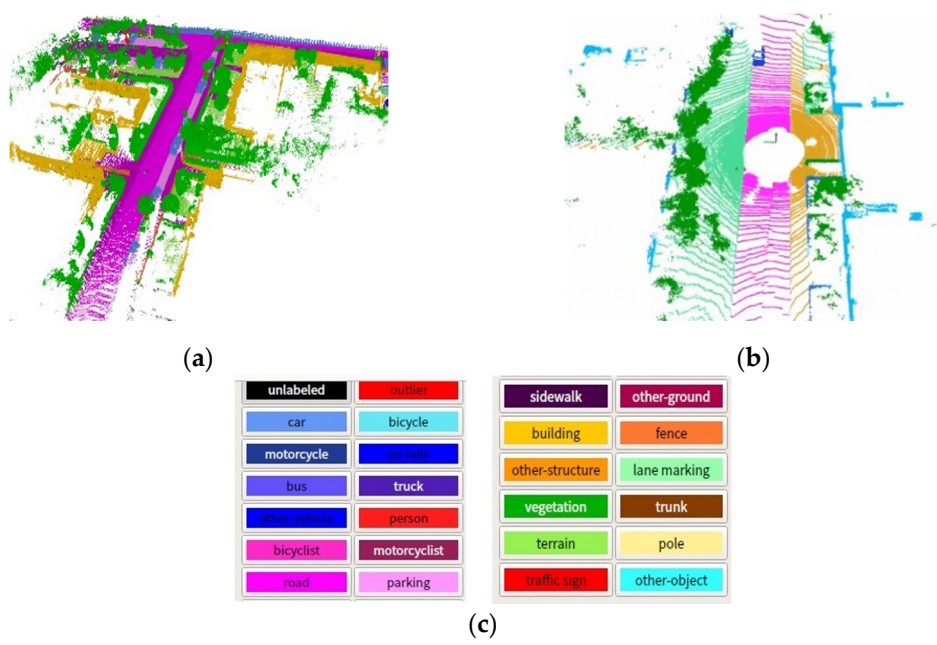

(**a**)                                                                  (**b**)

(**c**)

**Figure 17.** The comparison of point cloud semantic segmentation results: (**a**) segmentation visualization based on the improved RandLa-Net; (**b**) segmentation visualization based on the original RandLa-Net; and (**c**) the label information corresponding to the color.

**Table 4.** Test accuracy on semantickitti 03.

| Methods | RandLa-Net | Ours |
|:---:|:---:|:---:|
| mIoU(%) | 60.6 | 61.5 |
| parameter(M) | 1.24 | 1.24 |
| road | 92.1 | 92.3 |
| side-walk | 77.3 | 78.1 |
| parking | 42.6 | 43.9 |
| other-ground | 40.8 | 41.2 |
| building | 89.4 | 90.2 |
| car | 92.1 | 92.3 |
| truck | 50.8 | 51.3 |
| bicycle | 15.6 | 16.5 |
| motorcycle | 30.9 | 31.2 |
| other-vehicle | 41.7 | 41.5 |
| vegetation | 85.9 | 85.6 |
| trunk | 42.3 | 43.1 |
| terrain | 72.5 | 73.1 |
| person | 65.9 | 65.4 |
| bicyclist | 55.8 | 55.4 |
| motorcyclist | 8.9 | 9.7 |
| fence | 47.2 | 48.3 |
| pole | 53.1 | 53.3 |
| traffic sign | 40.4 | 41.7 |

From Table 4, it can be seen that the mIoU of all categories are improved after using the new method. This is because the shape features of each element on the global point cloud map are more complete, which is more conducive to be identified and extracted. However, in the classification of some small targets, the accuracy of the original algorithm is not as high as the frame-by-frame segmentation. This is because the registration of small targets in the registration time is not complete, which leads to some deviation in the subsequent classification.

### 5.6. Point Cloud Extraction

Since the file is in bin format, in order to find the point cloud data corresponding to the desired label from the classified point cloud map, the point cloud can be extracted according to the label. Figure 18a is the point extracted on label 15 cloud data, and Figure 18b is the point cloud data extracted on label 9.

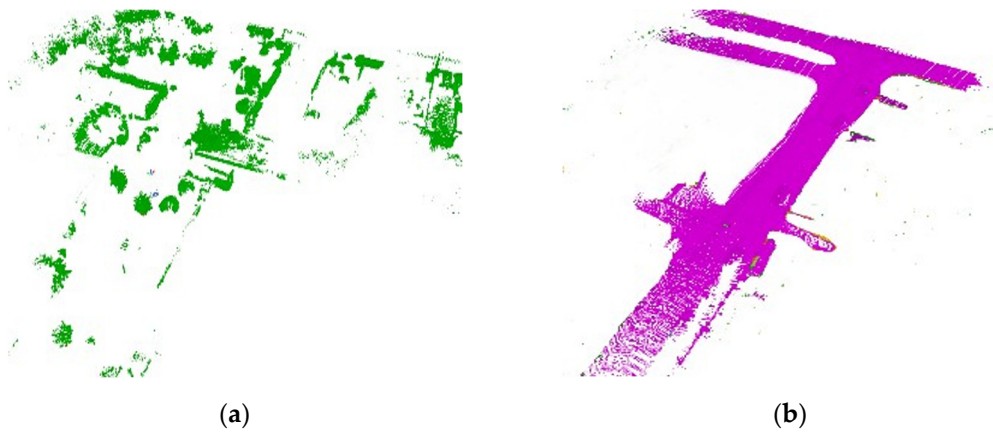

(**a**)                                             (**b**)

**Figure 18.** Point cloud extraction: (**a**) vegetation; and (**b**) road.

Figure 18 is the visual effect of extracting all the points marked vegetation and road in the SemanticKITTI dataset 03. It can also be converted into txt type and data in (x, y, z, label) format can be obtained for further operation. In addition to the two extraction examples of Figure 18, specific extractions can be performed based on 19 labels. The extracted data can be directly used to create high-definition maps after simple sorting. The sorted part of the data is shown in Table 5, where the category corresponding to the number is represented by a label.

**Table 5.** The form after data output.

| Numb | Object Type | Position Coordinates (x, y, z) | | |
|:---:|:---:|:---:|:---:|:---:|
| 1 | car | 20.354 | 40.375 | −2.404 |
| | | 20.356 | 40.374 | −2.404 |
| | | 20.359 | 40.374 | −2.399 |
| . . . | . . . | . . . | . . . | . . . |
| 6 | person | −0.847 | −34.686 | 3.215 |
| | | −0.852 | −34.683 | 3.212 |
| | | −0.852 | −34.683 | 3.215 |
| . . . | . . . | . . . | . . . | . . . |
| 9 | road | −0.003 | −31.752 | 0.002 |
| | | −0.001 | −31.756 | 0.002 |
| | | −0.002 | −31.759 | 0.002 |
| . . . | . . . | . . . | . . . | . . . |
| 15 | vegetation | −8.033 | −0.995 | −1.201 |
| | | −8.053 | −0.982 | −1.200 |
| | | −8.062 | −0.975 | −1.201 |

The data are labeled in the following order: 1. car; 2. bicycle; 3. motorcycle; 4. truck; 5. other-vehicle; 6. person; 7. bicyclist; 8. motorcyclist; 9. road; 10. parking; 11. sidewalk; 12. other-ground; 13. building; 14. fence; 15. vegetation; 16. trunk; 17. terrain; 18. pole; and 19. traffic sign.

### 6. Conclusions

This paper proposes a fast automatic classification and extraction method for large-scale point cloud data. The original point cloud data are globally mapped by NDT regis-

tration. In order to reduce the amount of computation, the point cloud map is randomly sampled. The cloud global map is semantically segmented, with each point assigned to a corresponding label, and then the numpy index is used to extract the point cloud data corresponding to the label of interest. The training results show that the point cloud data classification reaches 53.7% on the public dataset SemanticKITTI and 77.4% on Semantic3D. The test on the SemanticKITTI-03 dataset reflects that our method is more efficient than traditional manual annotation on large-scale point cloud datasets. However, in order to save computing power, the algorithm network parameters in the point cloud classification part are too few, resulting in an unsatisfactory classification effect on small target objects. In the follow-up research, the classification of small target objects will be further optimized and tested.

**Author Contributions:** Conceptualization, J.L. (Jiadi Li); methodology, J.L. (Jiadi Li); validation, J.L. (Jiadi Li); formal analysis, Z.M. and J.L. (Jiajia Liu); investigation, Y.Z., Y.W. and J.Z.; project administration, Z.M.; writing—original draft, J.L. (Jiadi Li); writing—review and editing, Z.M. and J.L. (Jiajia Liu) All authors have read and agreed to the published version of the manuscript.

**Funding:** This paper is supported by the International Cooperation Project of Science and Technology Bureau of Chengdu (no. 2019-GH02-00051-HZ), Sichuan Unmanned System-Intelligent Perception, Engineering Laboratory Open Fund, and the research fund of the Chengdu University of Information Engineering, under grant nos. WRXT2020-001, WRXT2020-002, WRXT2021-002 and KYTZ202142. This paper is also supported by the Sichuan Science and Technology program of China, grant no. 2022YFS0565.

**Institutional Review Board Statement:** Not applicable.

**Informed Consent Statement:** Not applicable.

**Data Availability Statement:** Not applicable.

**Acknowledgments:** The authors are grateful to the College of Automation, Chengdu University of Information Technology.

**Conflicts of Interest:** The authors declare that they have no conflict of interest to report regarding the present study.

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
