# Peer review of "An Improved RandLa-Net Algorithm Incorporated with NDT for Automatic Classification and Extraction of Raw Point Cloud Data"

_electronics, doi:10.3390/electronics11172795_

Round 1

Reviewer 1 Report

Unfortunately, this paper fails to provide sufficient discussions about the present development of the domain. The justification of this work is not strongly described and failed to point the key research problems this work tried to sovle. The contributions as claimed by the authors were not persuasive. The presentation is poor and difficult to follow. Some sentences are way too long. Not written in a serious scientific language style. The language should be careful proofread. Figures are not clear enough. The method and results were described without sufficient discussions. The significance of this work and generalization of the proposed method are not properly described. Literature cited in this work is not enough to cover this field. This work needs significant improvements to be reconsidered.

Author Response

Thank you for your comments and suggestions, which have given me a lot of inspiration. I have revised the article one by one according to your views. The specific content is placed in the word file.

Reviewer 2 Report

In my opinion, the paper -- although it should be interesting to the readers -- needs major revision. There are some reasons why is is necessary:

— In many places, there should be an explanation of abbreviations (many could be unknown to readers):

—— senmatic3D and senmatickitty in Abstract

—— RandLa-Net in line 60, etc.

— There are many typo errors:

—— "point; Finally" in line 20

—— "used When" in line 71

—— "to10Hz" in line 119

—— "Where" in line 289 (and there is bad indentation here as well)

—— "[CJ//" in more places in References, etc. (The work could be more careful…)

— The equations unusually use Roman letters for variables, e.g., x and y in (4). Normally, Italics are used for the variables. On the other hand, the Greek letters are written OK as phi in (4). Please revise the equations.

— Many figures are poor bitmaps, e.g., Figure 6, Figure 11 - this one looks exceptionally bad. It is necessary to use a vector-based picture editor to meet the Electronics journal standard…

— The algorithm at line 227 is also a bitmap, and the superscripts in the line 13 of this algorithm are practically unreadable. Why the algorithm is a bitmap copy from another software tool? The algorithms surely must be vector-based…

— The Lidar in Introduction should be defined a bit. (Line 27, not all readers are fully familiar.)

— I think the author's achievements (e.g., Tables 1 through 3) should be emphasized more clearly (and simply…).

— Figure 19(c) is unreadable.

— To Conclusion (line 409): is the global mapping absolutely reliable?

— And many errors in References, please check it.

Generally, the theme of the paper is actual (and important). However, the way how the paper was written should be improved considerably.

Author Response

Thank you for your comments and suggestions, which have given me a lot of inspiration. I have revised the article one by one according to your views. The specific content is placed in the word file。

Round 2

Reviewer 1 Report

Some concerns are addressed. But one of the major issues of this paper is that language failure. This is very disappointing. Careful proofread can identify the places and improve the manuscript significantly, however this does not happen. When I looked at the manuscript and the response letter, I can easily find numerous places to improve. Some mistakes are still out there without modifications.  Just name a few examples:

1) Line 286, you should use  comma symbols between '103', '104', and '105'. 

2) Figure 8 is not in a proper style. Please refer to other paper to see how to present algorithms

3) Figure 10 can be much better in a horizontal layout.

4) Figure 2 is unacceptable. Btw, if you mean 'No.' in the first column, you should not use 'Numb' which has another meaning!

5) In the newly added Line 29, 'Properly process the collected raw point cloud data. ', why this can be an independent sentence? 

6) Line 30, there is no space before 'As'. Is this acceptable?

7) Line 38, why 'Features' instead of 'features'?

Too many places are omitted and ignored by the author team. I strongly suggest the authors conduct thorough proofread before submissions. I see a failure of sufficient responsibility in these issues. I have to reject this work.

Author Response

Thank you very much for taking the time to read my manuscript and give us your valuable comments.

Reviewer 2 Report

The authors improved their paper according my comments and recommendations as written in coverletter.docx. Therefore, I recommend the paper for publishing now.

Author Response

Thank you very much for taking the time to read my manuscript and give us your valuable comments

Round 3

Reviewer 1 Report

In the response, for example, to the first point (2), the authors claimed that they revised. However, the manuscript was still not correctly revised.

'Line 30, there is no space before 'As'. Is this acceptable?'

Responded as 'In the manuscript, all the questions you mentioned have been corrected one by one.'

In this new version, Line 31, the space is still missing!

The language editing service is strongly recommended, such as 

https://www.mdpi.com/authors/english-editing 

Author Response

Thank you for taking the time to give me these valuable comments, and we are very sorry for our editing errors.
